# Determining the Dependence of a Landscape's Ecological Stability and the Intensity of Erosion during 1990–2018

Zuzana Németová *, Silvia Kohnová and Zuzana Sabová

Department of Land and Water Resources Management, Faculty of Civil Engineering, Slovak University of Technology in Bratislava, 811 07 Bratislava, Slovakia; silvia.kohnova@stuba.sk (S.K.); zuzana.sabova@stuba.sk (Z.S.)
* Correspondence: zuzana.nemetova@gmail.com; Tel.: +421-944512082

**Abstract:** Among the main elements that contribute to climate change are degradation processes and the ecological level of a landscape. These two topics have been discussed and researched for many years, and many studies have been conducted. The idea behind this article is to determine the correlation between the ecological stability of a territory and the intensity of degradation processes and find out how ecological stability affects the intensity of soil erosion and vice versa. The ecological stability was calculated based on various methods during the years analyzed, i.e., 1990, 2006, 2012, and 2018. The soil water erosion measurements were performed for the same period in order to identify the relationship between ecological stability and the intensity of soil erosion. The investigated area is located in the Slovak Republic, and each year reflects different types of management of the territory, reflecting the current situation in the catchment according to the year evaluated. The intensity of the erosion process was measured using a physically based EROSION-3D model based on the precipitation levels derived using the Community Land Model (the CLM). In addition to identifying the relationship between the level of ecological stability and the intensity of erosion, this study also describes the development of ecological stability during the evaluated period together with changes in soil erosion processes. The results show a dependence between the intensity of ecological stability and soil erosion. First of all, it determines whether such a dependence exists at all and also its extent.

**Keywords:** landscape ecological stability; erosion processes; EROSION-3D model





## 1. Introduction

The relationship between ecological stability and the degradation processes of landscapes is very close, and these two elements mutually influence each other. Areas in which there is a lower degree of ecological stability due to the action of various elements, i.e., inappropriate anthropogenic activity, natural threats, natural disasters, and the construction of buildings opposed to the environment's natural elements, are susceptible to various types of degradation processes. Maintaining environmental stability is also very desirable for agricultural landscapes since climate change also affects this sector [1] together with soil erosion, which poses a threat to maintaining a beneficial degree of stability and also poses a threat to crops [2]. Individual scientific concepts, i.e., ecological stability and soil erosion, have been examined on a relatively large scale, but their interconnection and effects on each other are still insufficiently clarified. On the other hand, knowing the relation between a low degree of ecological stability and an increasing degree of degradation could be beneficial for the successful identification of the areas affected and the subsequent implementation of conservation measures. Ecological stability is the ability of ecological systems (ecosystems) to resist negative external elements (natural and anthropogenic) through self-regulatory processes and their ability to return to their original state when the negative effects are over. External factors are considered to be stressors in a region that affect the natural evolution

of a landscape's ecosystem in a negative way and can often cause irreversible changes. It is about acting on the living organisms that create an ecosystem [3]. Ecological stability is also defined by the coefficient of ecological stability (CES), which is characterized by a numerical value and the corresponding interpretation of individual elements [4].

The following types of ecological stability have been distinguished [5]:

- Constancy means a minimal change in an ecological system or no change in an ecological system.
- Repetition in cycles defines changes in an ecological system that occur in regular cycles.
- The resistance of an ecological system characterizes the resilience of that ecological system to external influences or minimal changes due to external factors and the preservation of its structure to a certain extent.
- Resiliency means a change in an ecological system due to the action of an external factor and a return to the original state thanks to self-regulatory mechanisms.
- The dynamic balance of a landscape ensures the balance of fluctuations due to changing conditions, which result in a certain stability of the ecological system, which is manifested in its resistance to external disturbances. The opposite of this element is ecological lability.
- Ecological lability is a component without the ability to resist external factors, which consequently does not have the power to return to the original initial state of the ecological system.

The most commonly used principle for the assessment of ecological stability is based on two fundamental approaches:

1. The ecological stability coefficient is the ratio of the relatively stable and relatively unstable elements;
2. The ecological stability coefficient is determined on the basis of the acreage of landscape elements, taking into account the ecological significance of a landscape [6]. The connection between ecological stability and soil water erosion differences was defined based on the determination of soil water erosion processes during the years 1990, 2006, 2012, and 2018. The aim of the study was to identify if changes in ecological stability correlate with soil water erosion changes and how the intensity of erosion processes affects ecological stability. Both terms used (ecological stability and soil water erosion) are one of the most frequently analyzed terms in the scientific world because they are directly related to climate change. The sensitivity of soil water erosion can be defined in terms of land use and climate and is one of the most serious environmental degradation risks, negatively affecting the soil's attributes and functions as well. When we talk about the "degradation" of soil, we generally mean a process that decreases the current or potential capacity of the soil to provide a healthy basis for ecosystem services through human activities. Soil degradation is not only caused by a harsh climate; it can also be caused by poor land management practices and policies. According to Jankauskas (2003) [7], deforestation, a low level of landscape management, and overgrazing are the main reasons for the development of floods, wind, or water erosion. On the other hand, Borrelli (2020) [8] claims that inappropriate agricultural processes are the main reason for the generation of soil or environmental disruptions and that they also represent a major source of greenhouse gas emissions.

Long-term investigations into soil erosion necessitate knowledge of how climate change affects the geo-sustainability of ecosystems and their stability. The general relationship between climate and erosion is well known, and quite a few researchers have used it to make predictions about the likely correlation between climate change and soil erosion. As the temporal scale of interest has changed, the parameters of soil erosion models that have had fixed values for years have become unpredictable. Thresholds and discontinuities have emerged over longer time scales because of vegetation–soil interactions, human behavior, and the adaptive evolution of the environment under investigation. This means that as

research on a temporal or spatial scale expands, new mechanisms can arise that dominate soil erosion (e.g., an ecosystem disturbed by a wildfire) [9].

The relationship between biological diversity and ecological stability has inspired the interest of ecologists, but a growing understanding of the magnitude of human-caused climate change has prompted researchers to investigate whether and how ecological systems are able to withstand and recover from environmental stresses [10].

Modeling attempts (worldwide) to forecast the effects of climate and land use changes on soil are evolving, but they are constrained. Land use and, theoretically, climate change, through a more extreme hydrological cycle, are the two most important anthropogenic causes of erosion [8].

When analyzing landscape changes over time, specific land features, including their spatial representation, spatial configuration, and dynamics, are monitored. Research on territorial use, covering ancient times until the present, is critical for ecological studies that contribute to strategies for sustainable land use. Climate change, urbanization, deforestation, a loss of water quality, natural disasters, habitat destruction, and biodiversity loss all play a part [11].

The variety in the forms of stability that have been tested in theoretical and methodological research conflicts with its comparatively basic intuitive meaning. Understanding the equilibrium of ecological processes necessitates investigating the interdependencies and relative importance of these numerous facets of stability, i.e., various metrics, organizational scales, and perturbations. Indeed, what can be assessed in order to test the integrity of dynamic ecological systems? What is the scale? How does the stability scale across various organizational levels work? These are questions that remain largely unanswered [10].

This study reflects the evolution of ecological stability together with changes in the intensity of soil water erosion during the period covering the years 1990, 2006, 2012, 2018, and 2020. Each year, analyses of ecological stability according to several methods were made.

In this research, the following phenomena are discussed:

- The intensity of soil water erosion, together with ecological stability, was evaluated in order to reflect the relationship between changes in ecological stability and the water erosion of soil;
- A determination of the impact of changes in ecological stability on the intensity of water soil erosion;
- Reflections as to how the landscape changed during the selected years, i.e., 1990–2018, considering that these changes are projected in the degree of the ecological stability of the landscape as well as in the intensity of the erosion processes.

## 2. Materials and Methods

### 2.1. Study Area

The catchment under research is situated in the western part of Slovakia, close to the border with the Czech Republic (Figure 1). The area is a part of three protected landscape areas, i.e., the White Carpathians, the Little Carpathians, and Záhorie [12]. The geological conditions of the catchment area are represented by sandstone and sandy claystone in the upper part of the Teplica catchment area (in the White Carpathians, i.e., the Paleogene of the Carpathians). The lower part of the catchment is filled with calcareous siltstone, claystone, sandstone, tuffs, variegated and coal clay, coal, conglomerates, and organodetritic limestone. Fine-grained conglomerates and red claystone occur in the lower part of the Teplica river catchment area as well [13]. The annual amount of precipitation reaches a value of 600 mm. Most of the Teplica River catchment area belongs to zones of mildly warm and warm climates. On average, the moderately warm area has less than 50 summer days a year, with a maximum daily temperature above 25 °C and an average July temperature above 16 °C.

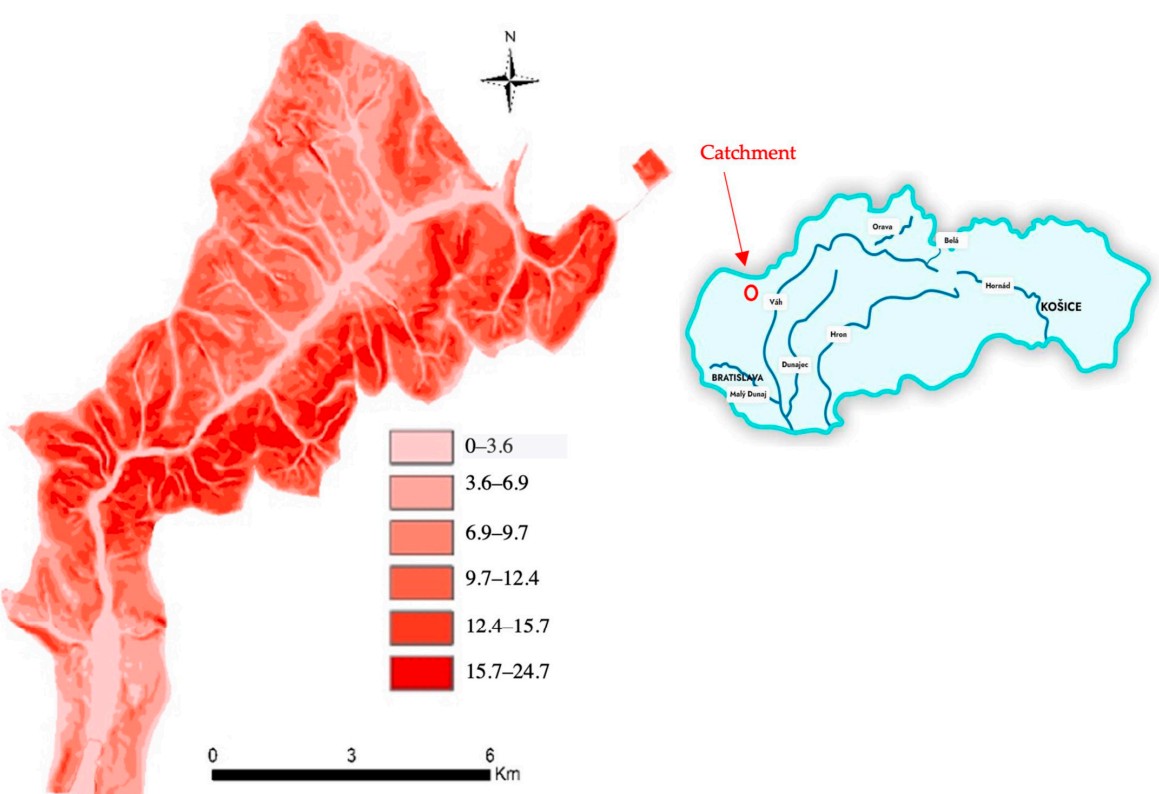

**Figure 1.** Slope representation and situation of the Teplica River catchment area [12].

### 2.2. Characteristics of the Input Data

The soil water erosion and the coefficients of ecological stability were measured for the years 1990–2018. For this purpose, a diagram describing the methodological procedures of the research was created and is visible in Figure 2. The rainfall data were derived using the Community Land Model (CLM), which is described in Section 2.4. A graphical interpretation of the rainfall amounts received is displayed in Figure 3. A detailed identification of the individual landscape elements with their percentual representation is contained in Figure 4. It is obvious that more than 50% of the territory is covered by agricultural land every year. The Community Land Model (CLM) was used to estimate the real total precipitation for the river basins analyzed, as no real observed rain gauge data are available for them. A ten-minute step precipitation was used for the selected periods, i.e., 1990, 2006, 2012, and 2018. Figure 3 shows the monthly precipitation totals to indicate the precipitation rates of the individual months. The determination of soil erosion was carried out using the physically based EROSION-3D model with the input data described above (Figure 5). The characteristics of the EROSION-3D model are described in Section 2.5. Since land use management plays an important role in the processes of soil erosion and other soil processes associated with it, the modelling of water erosion was performed for the years selected with the land-use structure composition (land cover) shown in Figure 6A–D.

### 2.3. Analyses of Ecological Stability

The following qualitative coefficients of ecological stability were developed for the conditions in Slovakia and used to determine the coefficient of ecological stability:

The coefficient of ecological stability, according to [14], is as follows:

$$CES = \frac{S}{L} \tag{1}$$

where S is an area of relatively stable land (forests, non-forest woody vegetation, meadows, pastures), and L is an area of relatively unstable land (arable land, built-up land).

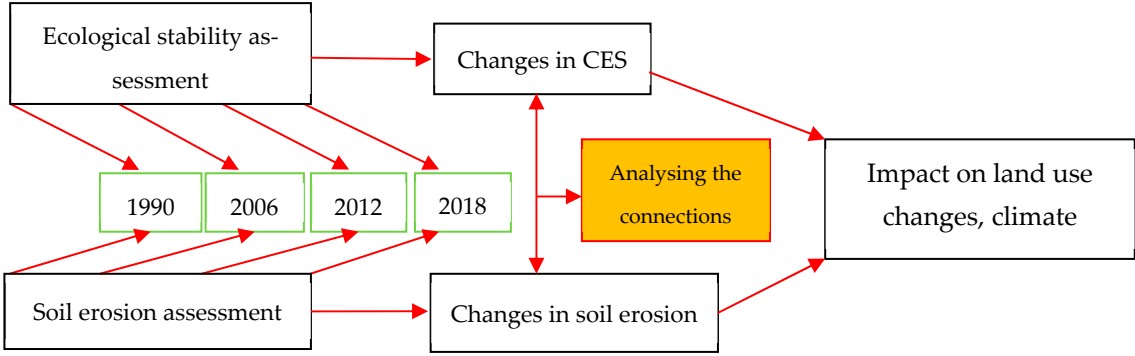

**Figure 2.** The scheme of the procedure and the continuity of the assessment of ecological stability and soil erosion intensity within the study.

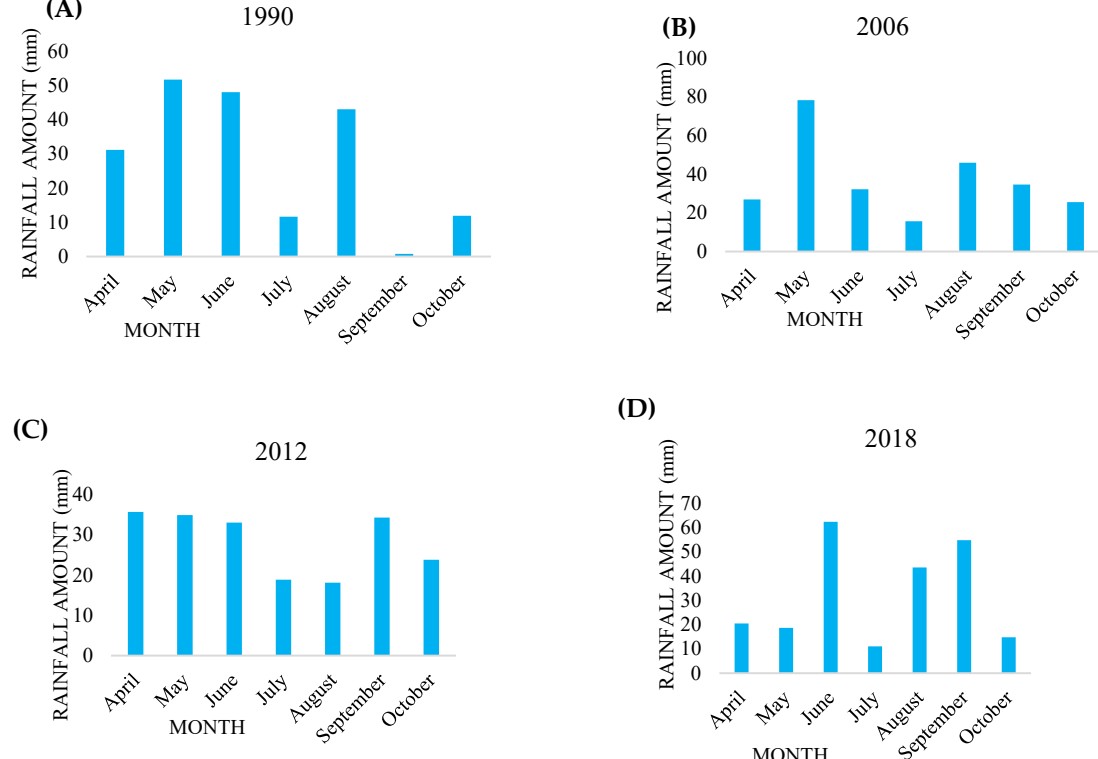

**Figure 3.** Monthly amounts of precipitation events derived from the Community Land Model (CLM model): (**A**) 1990, (**B**) 2006, (**C**) 2012, and (**D**) 2018.

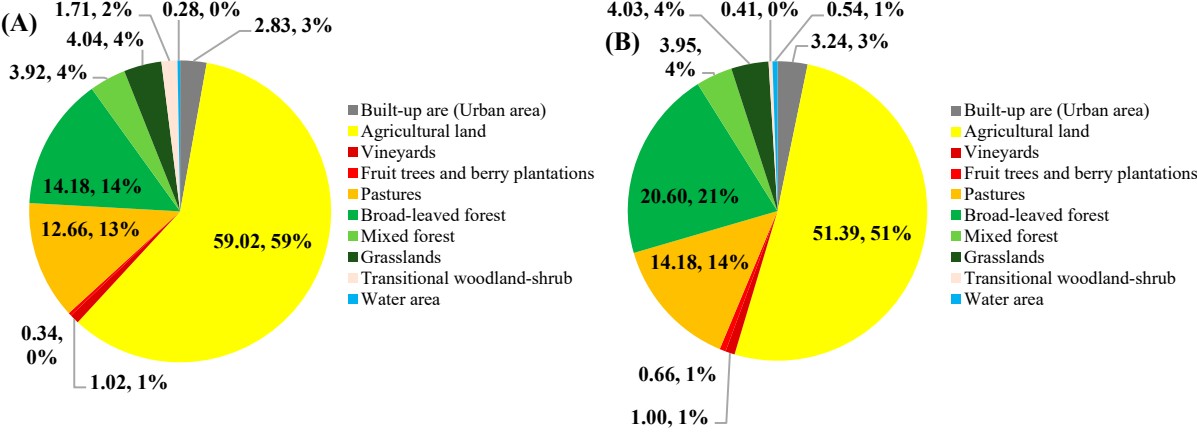

**Figure 4.** *Cont.*

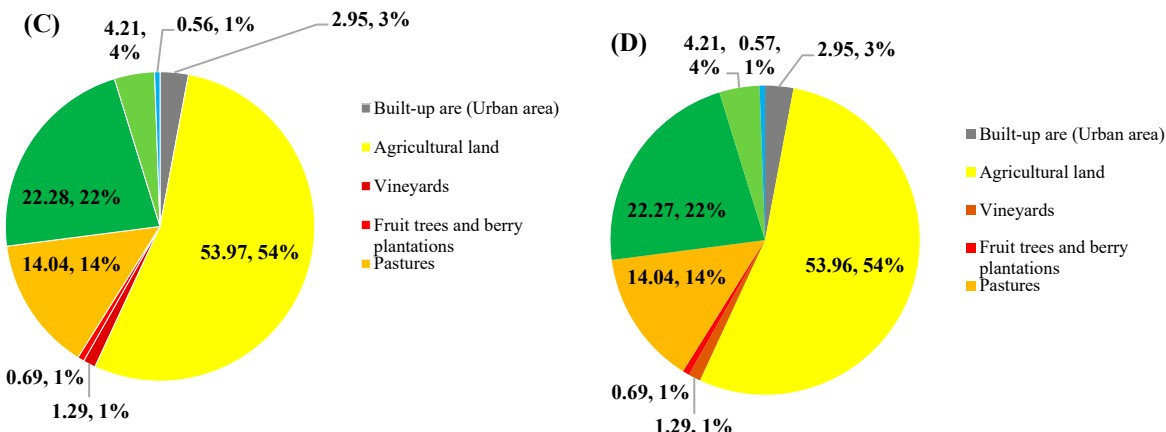

**Figure 4.** Percentual and graphical representation of individual elements of the land use structure for the selected years: (**A**) 1990, (**B**) 2006, (**C**) 2012, and (**D**) 2018.

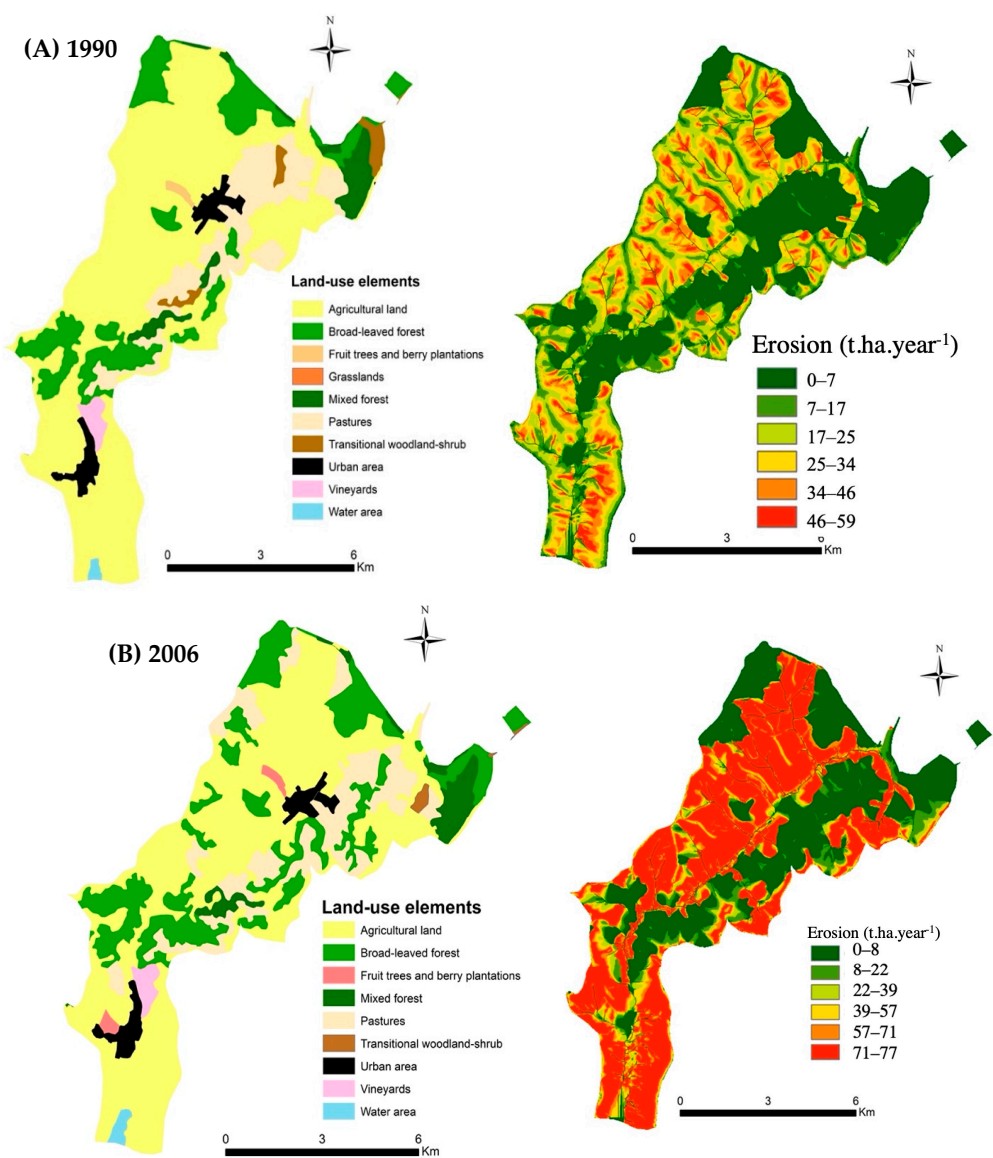

**Figure 5.** *Cont.*

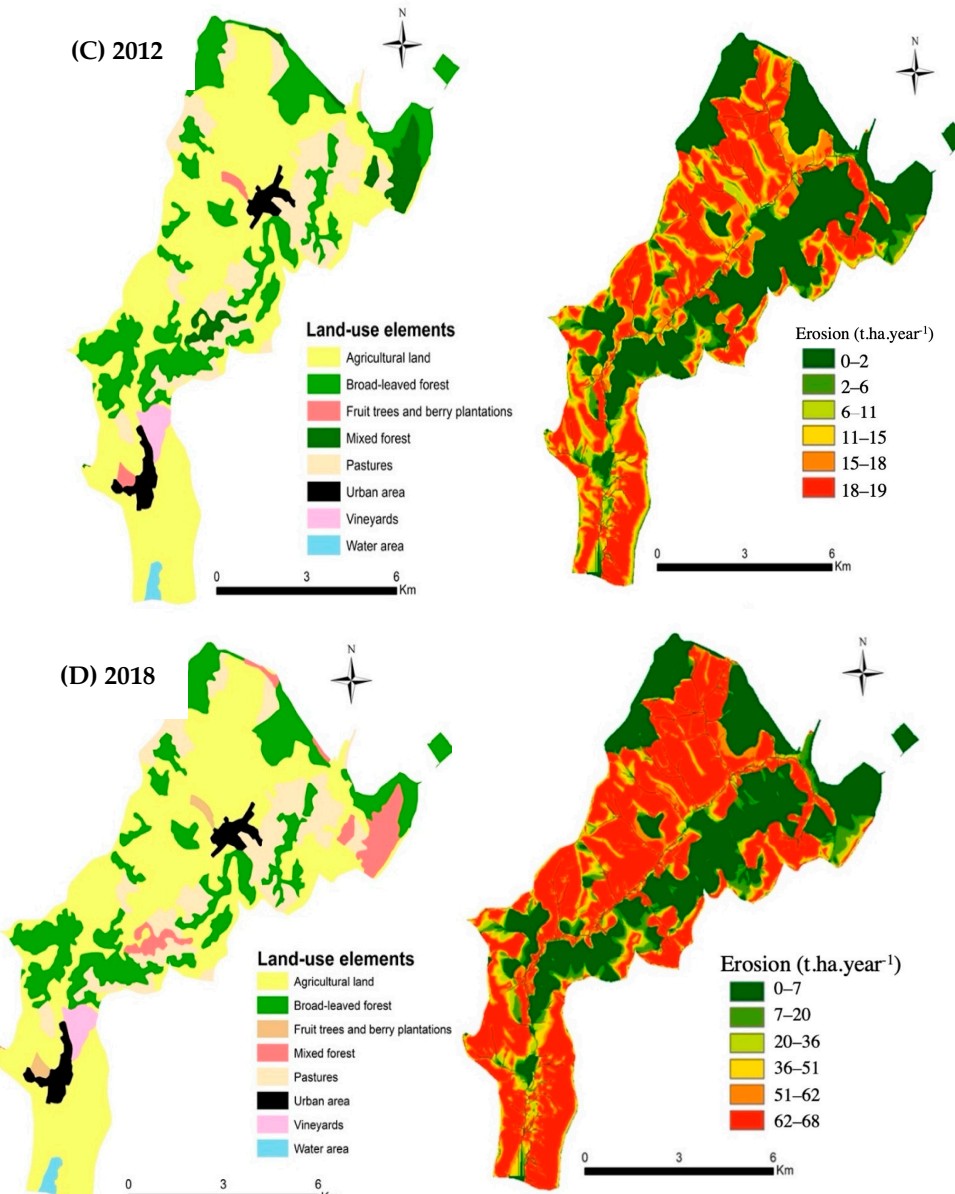

**Figure 5.** (**A**–**D**). Land use structure and net erosion; scenario C (output EROSION-3D model with editing in GIS 2020).

*An evaluation of the stability of the landscape according to the methodology presented in* [14]:

- A CES < 0.10 indicates territory with maximal disturbance to the natural structures; the basic ecological functions must be intensively and permanently replaced through technical interventions;
- A CES in the range of 0.10–0.30 indicates a territory with intensive use featuring a clear disruption to natural structures;
- A CES in the range of 0.30–1.00 indicates a territory intensively used mainly for agricultural production; a large number of self-regulatory processes cause ecological lability;
- A CES > 1.00 indicates an almost balanced landscape in which the technical objects are relatively in harmony with the natural structures

The coefficient of ecological stability according to [15]:

$$Cs = \sum_{i=1}^{n} \frac{pi.cpi}{p} \qquad (2)$$

where Cs is the coefficient of the cadastral area structure; Pi is the area of the individual elements of the landscape in hectares; Cpi is the coefficient of the ecological significance of the elements; P is the size of the cadastral area in hectares; and N represents the number of elements in the cadastral area.

*An evaluation of landscape stability according to the methodology in* [15]:

- CES < 0.30—the poorest landscape structure;
- CES 0.31–0.40—a poor landscape structure;
- CES 0.41–0.50—a low-quality landscape structure;
- CES 0.51–0.60—a moderately high-quality landscape structure;
- CES 0.61–0.70—a medium-quality landscape structure;
- CES 0.71–0.80—a significantly high-quality landscape structure;
- CES > 0.81—the best landscape structure.

The coefficient of ecological stability according to [16]:

$$CES = \frac{(\ AL + UA + OA\ )}{(\ Ga + Gr + Vi + Or + Fo + WA\ )} \tag{3}$$

where *CES* is the coefficient of the ecological stability of the area; *AL* represents arable land [ha]; *UA* is an urban area (built-up area) [ha]; *OA* is another area [ha]; *Ga* refers to gardens [ha]; *Gr* is permanent grasslands [ha]; *Vi* is vineyards [ha]; *Or* is orchards [ha]; *Fo* are forest areas [ha]; and *WA* represents water areas [ha].

*An evaluation of landscape stability according to the methodology in* [16]:

- CES < 1.00—predominantly natural landscape elements;
- CES = 1.00—a balanced landscape;
- CES > 1.00—predominantly anthropogenic landscape elements.

The coefficient of ecological stability according to [6]:

$$CES = \sum_{1}^{n} \frac{pi.Si}{p} \tag{4}$$

where *CES* is the coefficient of the ecological stability of the area of interest; *pi* is the total size of the individual types of elements of a landscape structure (ha); *Si* is the degree of ecological stability; *P* is the total size of the area of interest (ha); and *n* is the number of elements of the landscape structure in the area of interest.

*An evaluation of landscape stability according to the methodology in* [6]:

- CES 1.00–1.49—a landscape with very low ecological stability;
- CES 1.50–2.49—a landscape with low ecological stability;
- CES 2.50–3.49—a landscape with medium ecological stability;
- CES 3.50–4.49—a landscape with high ecological stability;
- CES 4.50–5.00—a landscape with very high ecological stability.

## 2.4. CLM Model

The Community Land Model (CLM) is a part of the Community Earth System Model (CESM), and the model describes the ideas behind ecological climatology. The major aim of the model is to determine the impact of natural and human factors on the climate. The model contains submodels related to land biogeophysics, the hydrologic cycle, biogeochemistry, human dimensions, ecosystem dynamics, and components of land surfaces containing surface heterogeneity [17]. The model was developed with the aim of researching the physical, chemical, and biological processes by which terrestrial ecosystems impact and are impacted by climate on spatial and temporal scales [17].

## 2.5. EROSION-3D Model

The physically based EROSION-3D model is an event-based method with the main goal of determining the amount of surface runoff, the soil loss and sediment generation,

and the deposition processes caused by intensive precipitation rainfall events. This model is mainly used for agricultural land and describes the erosion process in a complex way. That is why it is composed of two submodels, i.e., the infiltration and erosion submodels. The erosion submodel describes soil erosion processes in three steps, i.e., the detachment of soil particles by the impact of raindrops, their transport, and their deposition. This submodel includes the generation of surface runoff and the detachment of soil particles through the kinetic energy of raindrops and surface runoff. The mathematical expression of the erosion submodel is based on the momentum flux approach defined by the following equation [18]:

$$\varphi_q D = \frac{q * \rho_q * v_q}{\Delta x} \tag{5}$$

where $\varphi_q D$ is the momentum of the flux exerted by an overland flow; q is the flow [$m^3/(m*s)$]; $\rho_q$ is the fluid density [$kg/m^3$]; $v_q$ is the mean velocity of the flow according to Manning [m/s]; and $\Delta x$ is the length of a specified slope segment [m].

The infiltration submodel uses the Green–Ampt approach to define the process of infiltration and considers the soil as a rigid and homogeneous soil matrix (vertical changes in the physical soil properties, dynamic processes, or changes in soil structure due to biological activity are not considered) [19].

A mathematical description of the infiltration process includes the gravitational component $i_1$ and the dynamic component of the matrix $i_2$.

The gravitational potential is defined as a function of the gravitational component ($i_1$):

$$i_1 = k \cdot \frac{\Delta\psi_g}{x_{f1}} = k \cdot g \tag{6}$$

where $i_1$ is the infiltration rate of the gravitational component [$kg/(m^2.s)$]; k is the hydraulic conductivity of the transport zone [$(kg.s)/m^3$]; $\Delta\psi_g$ is the gravitational potential [$(N.m)/kg$]; $x_{f1}$ is the depth of the wetting front of gravitational component [m]; and g is the gravitational constant [$m/s^2$].

The matrix potential, $\psi m$, is described by the matrix component $i_2$:

$$i_2 = k \cdot \frac{\Delta\psi_m}{x_{f_2}(t)} \tag{7}$$

where $i_2$ is the infiltration rate of the matrix component [$kg/(m^2.s)$]; k is the hydraulic conductivity of the transport zone [$(kg.s)/m^3$]; $\Delta\psi_m$ is the matrix potential [$(N.m)/kg$]; and $x_{f2}(t)$ is the depth of the wetting front of the gravitational component [m] at time t.

The saturated hydraulic conductivity is defined by an empirical equation according to [17] and depends on the soil structure, soil texture, and the presence of macrospores.

As mentioned previously, the EROSION-3D model was predominantly established as an event-based model, but a submodel has been developed that can now perform long-term simulations. Consequently, the long-term simulation is based on a continuous rainfall series consisting of a series of single rainfall events that occurred within the period evaluated. Each rainfall event requires its own soil data set, the parameters of which account for the individual soil conditions and the stages of crop growth as of that date.

The EROSION-3D model can be used for the calculations of soil loss, the detachment of soil particles, the transport and deposition of detached soil particles, and natural or artificial rainfall events. The model is physically based, which means that it was primarily established on physical principles and mathematical equations. The basis of the model's principles is built on the momentum flux approach evolved by [18]. Since the model was developed to predominantly simulate single rainfall events (an event-based model), a long-term simulation submodel was constructed in order to perform long-term simulations. The arrows show the individual parameters that are entered into the equations and which participate in the model processes.

## 3. Results

The aim of the article is to evaluate the relationship (connection) between ecological stability and the intensity of soil water erosion during the period established. The goal of the study was to find the correlation between the intensity of soil water erosion processes and the ecological stability of the research area. The study includes several models and methods that have been chosen and applied, i.e., the physically based EROSION-3D model, the Community Land Model (the CLM model), and methods for assessing the coefficient of ecological stability. The purpose of the study lies in finding and analyzing the relationship between ecological stability and the intensity of soil water erosion within the selected years.

The results summarize the achievement of the following objectives:

(a) An analysis of the development of ecological stability for the years selected, i.e., 1990, 2006, 2012, and 2018;

(b) A determination of the changes in the landscape in terms of the distribution of individual landscape elements and the extent to which these changes have affected ecological stability and erosion intensity;

(c) An estimation of the relationship between ecological stability and erosion processes, i.e., defining the connection and determining its dependence.

The development of ecological stability was analyzed using ecological stability co-efficients according to different methodologies. Four methodical principles for assessing ecological stability were used. The methods selected are based on different approaches developed by various authors. The ecological evaluation of the landscape of the Teplica River catchment area took place, where the gradual development of the territory and changing areas of the landscape elements were monitored. The positive ecological elements have had an increase of 7% since 1990 compared to the negative elements. These include the following landscape elements: deciduous and mixed forests, meadows and tall grass, orchards, plantations, pastures and low grasses, transitional forest cover, vineyards, and water areas in the area analyzed. The forests represent 29% of the land. The negative (unstable) ecological components include agricultural and urbanized areas. From 2006 to 2018, we did not observe any percentual changes in the elements. The least favorable conditions in the area were monitored during 1990; since then, the positive element values have started to increase. They have reached 43%, but the negative elements still prevail. Nowadays, agricultural landfills constitute the majority of the landscape (more than half of the land). A summary of the assessment of ecological stability by different methods can be found in Table 1, where the results from the methods used as well as the results from the EROSION-3D model are displayed. A summary of the landscape elements with information about the area is contained in Table 2. The percentual representation of the individual elements (positive and negative) shows that since 2016, more than half of the area (57%) has been covered by positive elements. Figure 6 displays the relationship between the intensity of an erosion and the ecological stability coefficient according to various authors [6,14,15,17]. A graphical interpretation of the development of ecological stability is shown in Figure 7. Based on this graph, it is clearly visible which methods achieved which values within the individual years. The land use situation that occurred during the years 1990, 2006, 2012, and 2018 is shown in Figure 6A–D, together with the intensity of erosion related to that land use composition and those years as well. Significant manifestations of soil water erosion were determined in most of the years, but the most intense was found in 1990. The reason for this is the amount of arable land covering the territory in the year 1990 (59% of arable land). The relationship between the development of the intensity of soil erosion and rainfall events is displayed in Figure 8. According to the ecological stability coefficients, a low ecological coefficient of ecological stability was calculated for all the methods used, and thus the area is considered an intensively used territory with poor quality and a dominance of anthropogenic landscape elements. Regarding the relationship between the ecological stability coefficients and the erosion processes, in most cases, a connection was found. As the rate of erosion increased, the ecological stability of the territory decreased simultaneously, and vice versa. When investigating the Erosion-3D

model, the significance of precipitation was identified as the biggest factor determining the model's result.

Many studies address the importance of the amount of rainfall as an important element involved in the generation of erosion processes with an upward trend due to climate change [20–23]. In some cases, the intensity of soil erosion may differ independently of the ecological stability values because the ecological stability coefficient does not consider precipitation. In the study, the diversity of this characteristic can be seen in the year 2018, when the methodology from [4] was used, and where, with an increase in the ecological stability indicator, the intensity of erosion processes also increased. The connection between the intensity of soil erosion and ecological stability is evolving in the sense of an indirect ratio, which means that by increasing the value of ecological stability, the intensity of erosion processes is reduced. According to this trend, the assumption about the impact and the link between ecological stability and erosion intensity can be confirmed. This fact may be particularly useful in areas that are already affected by some degree of soil erosion. In the case of such regions, it is necessary to carry out remediation, which will help prevent ongoing erosion processes and minimize their impact.

**Table 1.** A summary of the assessment of ecological stability by different methods and according to the intensity of soil erosion.

| Methods of CES Assessment by Authors | Time Period | CES (Coefficient of Ecological Stability) | Evaluation of Landscape Management According to CES | Intensity of Erosion * (t/ha/year) (EROSION-3D Model) |
|---|---|---|---|---|
| Míchal [12] | 1990 | 0.55 | Intensively used territory | 60.77 |
| | 2006 | 0.63 | | 65.30 |
| | 2012 | 0.76 | | 49.43 |
| | 2018 | 0.66 | | 64.71 |
| Miklós [13] | 1990 | 0.39 | Poor quality | 60.77 |
| | 2006 | 0.42 | | 65.30 |
| | 2012 | 0.56 | | 49.43 |
| | 2018 | 0.46 | | 64.71 |
| Kupková [14] | 1990 | 1.11 | Predominance of anthropogenic landscape elements | 60.77 |
| | 2006 | 1.31 | | 65.30 |
| | 2012 | 1.52 | | 49.43 |
| | 2018 | 1.14 | | 64.71 |
| Reháčková, Pauditšová [4] | 1990 | 1.86 | Landscape with low ecological stability | 60.77 |
| | 2006 | 1.98 | | 65.30 |
| | 2012 | 2.08 | | 49.43 |
| | 2018 | 2.01 | | 64.71 |

Note: * the values represent the results of the model's output, called "net erosion".

**Table 2.** Summary of the ecological evaluation of the landscape of the Teplice territory.

| Landscape Elements | Year | | | |
|---|---|---|---|---|
| | 1990 | 2006 | 2012 | 2018 |
| | Area [ha] | | | |
| Deciduous forests | 1130.33 | 1647.80 | 1704.04 | 1704.04 |
| Meadows, tall grass | 0.03 | - | - | - |
| Orchards and plantations | 27.32 | 52.87 | 52.87 | 52.87 |
| Pastures, low grass | 1009.54 | 1133.71 | 1074.18 | 1074.18 |
| Agricultural land | 4704.58 | 4109.57 | 4129.12 | 4129.12 |
| Transitional forest cover | 136.42 | 33.32 | - | - |
| Urbanized area | 225.98 | 233.13 | 226.35 | 226.35 |
| Vineyards | 81.03 | 80.46 | 98.55 | 98.55 |
| Water areas | 22.37 | 43.42 | 43.14 | 43.14 |
| Mixed forests | 312.94 | 316.25 | 322.27 | 322.27 |

According to Figure 6, the intensity of erosion processes reaches its highest peak in 2006 in comparison with the year 2012, where the lowest intensity of erosion was modeled. This phenomenon can be explained by the precipitation rate in the year 2006, when the highest volume of precipitation was recorded. When comparing the intensity of erosion and the coefficient of ecological stability in 2012, a decreasing trend in erosion can be observed with an increase in ecological stability compared to 2018, where the intensity of erosion declined with a decreasing coefficient of ecological stability. On the contrary, in 1990 and 2006, there were no significant relationships observed between the erosion intensity and correlation coefficients.

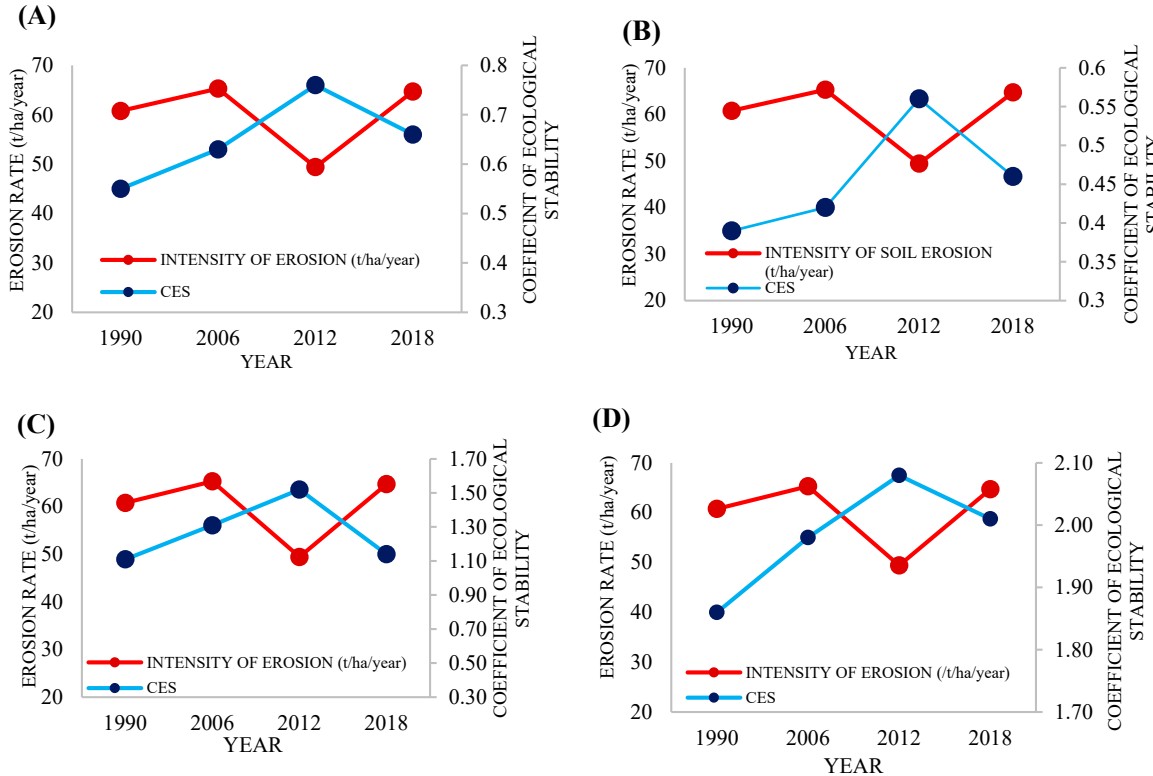

**Figure 6.** The relationship between the intensity of erosion and the ecological stability coefficient according to various authors: (**A**) Míchal (1982), (**B**) Miklós (1986), (**C**) Kupková (2001), (**D**) Reháčková, and Pauditšová (2007).

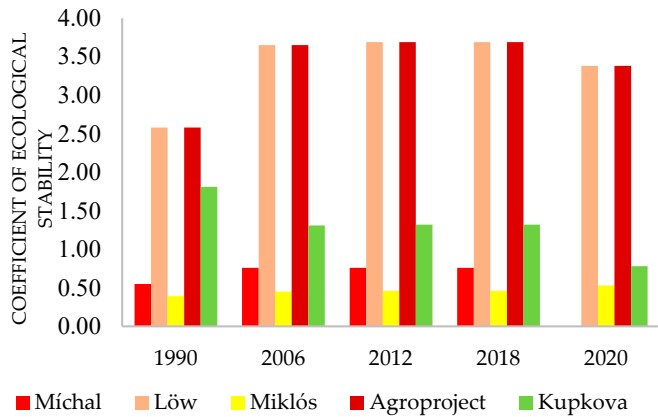

**Figure 7.** The summary of the assessment of ecological stability in the selected period (1990–2020) according to the methodologies used.

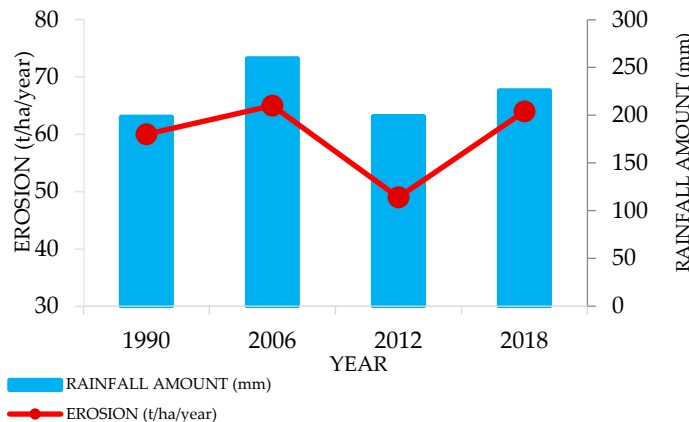

**Figure 8.** The connection between the rainfall amount (mm) and the intensity of erosion (t/ha/year).

## 4. Discussion

We live in a time when we are witnessing the mass construction of buildings and a reshaping of landscapes according to our needs, whether it is for recreational purposes or for residential or industrial purposes. Although they are not always negative interventions, e.g., in the case of modifications for parks, where the goal is primarily to create a harmonious environment, even so, they will always affect the landscape's original state in a certain way (e.g., in the case of animals that will have to adapt to new living conditions).

Every inappropriate intervention in a landscape can contribute to the extension of climate change and to changes in a landscape´s pattern, hydrological cycle, or ecological principles due to the effects of climate change. With the growing human population and developmental and technological progress, landscapes are exposed to constant changes and pressure. These changes can have a positive as well as a negative effect, depending on the purpose for which they are made. For instance, land consolidation can optimize and increase soil conditions (through the construction of anti-erosion and water management facilities, fields, and forest roads) and upgrade agricultural management practices as well [24]. On the other hand, activities such as urbanization, rock removal, deforestation, uncontrolled quarrying, unsuitable land use management practices, and the mass construction of human residences and factories contribute to climate change and to changes that man cannot even estimate. Predicting the future development of the water erosion of the soil caused by climate change or landscape change is a very difficult and time-demanding process, and there are still not enough methodologies or research methods to deal with this topic. Although many studies were conducted to consider landscape transformations [25–27], the works examining the impacts of both elements, i.e., climate change and land use transformations, are inadequate. This also means a reduction in $CO_2$ emissions through land use since land use and cover are two of the factors most contributing to $CO_2$ emissions [28] and also influence, to a large extent, biological cycles and ecosystem services [26]. According to Sulaeman (2020) [29], the best technical measure to prevent soil erosion consists of suitable agricultural practices, support for land management, and the recovery of places that are already affected by soil erosion.

However, it is important to mention that there are some environments that are inherently unstable without the influence of any external factors [30]. The causes of this instability may lie in internal factors such as trophic behavior and demographic stochasticity [31]. The connection between ecological stability and soil water erosion is mutual, i.e., there are about 75 billion tons of soil eroded from the world's terrestrial ecosystems. The rate of soil loss from agricultural land ranges from 3 tons/ha/year to 40 tons/ha/year. Since the rate of soil renewal is much slower, the soil is being lost 13–40 times faster than it is being formed [32]. We cannot ignore the fact that erosion worsens the quality of land and has a significant effect on people, the environment, wildlife, industry, and the economy [33].

Many studies have been performed that reflect the consequences of inappropriate anthropogenic activities and their effects on landscapes [27,34–40]. One of the consequences caused by unsuitable anthropogenic activities is land degradation (in various forms), which is a major problem worldwide. Among all degradation processes, the water erosion of soil is the most serious problem. In many parts of the world, irreversible soil losses have occurred that represent a significant environmental threat to sustainability with a direct impact on climate change and food production. The connection between soil water erosion and climate change is well known, especially due to climate-warming carbon dioxide, the emission of greenhouse gases (GHGs), and an increased amount of carbon uptake, considering that it is not possible to grow plants on degraded soil. Lastly, it is desirable to mention that by increasing the ecological stability of an already damaged territory, land degradation processes tend to decrease. Because land degradation directly impacts climate change, by increasing the overall prosperity of a landscape´s ecological stability, a potential reduction in the adverse effects of soil degradation on climate change may be achieved.

## 5. Conclusions

The aim of this article is to identify the correlations between the ecological stability of a territory and its soil erosion intensity in the years 1990, 2006, 2012, and 2018. The study was conducted for land use related to this specific period.

The mutual relationship between the elements described above was determined to be that by increasing the coefficient of ecological stability, the intensity of erosion processes decreases. This trend was confirmed in 88% of cases in the years analyzed, which can be concluded as showing a relatively high dependence and connection between the ecological stability of the area and the intensity of soil erosion. The article delineates the impact of changes in land use and the ecological stability of the territory and provides an ecological evaluation of the landscape and climate change. A link between the ecological stability of the territory and the intensity of soil erosion was observed and analyzed. Based on the range of years selected, changes in ecological stability and the intensity of soil water erosion were observed. The individual coefficients of ecological stability differ minimally from each other, and the area evaluated is considered to be intensively used with a poor quality of landscape management and low ecological stability. The intensity of erosion processes was defined as beyond the permissible limit in all the years evaluated. This means that a reduced rate of ecological stability is correlated with an increased rate of erosion, which was the main finding of the article. The importance of this study can be seen in the significance of maintaining favorable ecological stability through positive changes in the landscape and, thus, reducing and protecting against the creation of soil degradation processes. It can be said that the land, which has favorable ecological stability, is less susceptible to soil erosion processes, and soil relations and the overall health of the land are preserved.

**Author Contributions:** Design of the research paper, Z.N. and S.K.; collection of the observed data, Z.N.; formal analysis, Z.N.; methodology, Z.N. and S.K.; writing the original draft, Z.N.; providing critical points for further improvement of the manuscript, S.K. and Z.S.; writing—review and editing, Z.N. and S.K.; funding acquisition, S.K. All authors have read and agreed to the published version of the manuscript.

**Funding:** This research was supported by the Slovak Research and Development Agency under Contract No. APVV 19-0340 and the VEGA Grant Agency No. VEGA 1/0782/21. The authors are grateful for the support of their research.

**Data Availability Statement:** Data are contained within the article.

**Acknowledgments:** We sincerely thank the editor and two reviewers for their helpful comments, which significantly improved the quality of the manuscript.

**Conflicts of Interest:** The authors declare no conflicts of interest.

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
