# Peer review of "Determining the Dependence of a Landscape’s Ecological Stability and the Intensity of Erosion during 1990–2018"

_water, doi:10.3390/w16030378_

Round 1

Reviewer 1 Report (New Reviewer)

Comments and Suggestions for Authors

Overall, the problem presented in the manuscript is very interesting. However, I believe it should be improved for publication.

Title: I suggest changing it because it contains a contradiction - development, ecological condition and erosive intensity

Abstract: for reconstruction; lack of research purpose, too much methodological information; unconvincing results and conclusions for a summary

Introduction: please quote according to whom the information is provided, lines 86, 88, 236, 540; please do not repeat information in the discussion

Material and methods: please keep the diagram and do not provide graphic information without comments; why precipitation was analyzed only for 7 months of the year; Fig 4 missing letter a in the word area; Giving the years 1990-2018, we suggest research only in these years - line194; line 200 (if not the most important) is not a scientific statement; line 220-225 I think it should be CES, not KES;

For what purpose were so many indicators used, provide the justification for their use;

line 229 what criterion is it based on

line 283 rainfall events - maybe another word

Results: in my opinion, the form of presenting the results is not appropriate; There is an introduction and purpose, as well as elements of discussion with citations - we do not do this in this chapter; some information is repeated, e.g. lines 343-349. No description of raw results. Because it is not only about informing that it is presented in the form of a chart or table. Please describe these results. There is no statistical analysis of these results, e.g. correlations. Fig. 6 since there are such small differences, this information can be provided in the text.

Discussion: The discussion presented does not refer to the results obtained in this work. Rather, it may serve as an introduction.

Personification should be avoided - line 496

Conclusions: These are not conclusions from research undertaken in the work, in my opinion it is an abstract

Author Response

Dear reviewer,

you will find answers to your comments in the attached document.

Reviewer 2 Report (New Reviewer)

Comments and Suggestions for Authors

In this article, the authors identify determining factors of climate change, referring to the degradation processes and the ecological level of the landscape. In this way they were able to correctly correlate them, which results in an important advance in this field of knowledge. Likewise, it was possible to know the effect between ecological stability and soil erosion, important elements in the analysis of degradation processes.

At the same time, we may require some adjustments and pressures that we will list:

1. Figure 1 requires greater breadth, showing at least which continent it is located on, some nearby natural elements such as important rivers, lagoons, lakes, elevations, among others.

2. It is advisable to incorporate more information regarding the climate in the study area, especially precipitation, at least one extreme hyetograph. When it comes to soil erosion processes, it is important to accurately have the precipitation intensity process in the area.

3. Figure 6. Summary of the graphic interpretation of the positive and negative elements of the Teplice catchment (1990, 2006, 2012, and 2018), contains information that can be described, it is not very relevant to be represented in a figure, I suggest be eliminated.

4. In reality, the figures as a whole are not very relevant, it would be recommended that the authors reformulate the presentation of results and look for a more interesting graphic form that expresses the presentation of results with respect to the stated objective.

5. We recommend that the discussion of results be improved, including recent references.

Author Response

Dear reviewer,

you will find answers to your comments in the attached document.

Round 2

Reviewer 1 Report (New Reviewer)

Comments and Suggestions for Authors

Thank you very much for responding to your comments and I accept the work in its current form.

Author Response

Dear Reviewer,

thank you very much for your comments and time devoted to the article.

Best regards

The authors

Reviewer 2 Report (New Reviewer)

Comments and Suggestions for Authors

The article was improved according to the previous revision but still requires some corrections such as:

- To improve the discussion of results, it is recommended to use some references from similar cases such as (Condori-Tintaya et. al., Soil loss due to water erosion on semi-arid slopes of the Cairani-Camilaca sub-basin, Peru Idesia 2022, 40 (2), 7-15. https://dx.doi.org/10.4067/S0718-34292022000200007)

- The representation of results in a pye-type diagram in Figure 6 is of little importance, I suggest it be eliminated and a description of this data be made.

- On line 522, a non-coherent paragraph is presented, it is isolated, an exhaustive review of the wording of the article is suggested

- There are still formatting errors in the reference, review carefully.

Author Response

Dear Reviewer,

thank you very much for your comments and time devoted to the article. The answers are inserted in the attached document.

Best regards

The authors

This manuscript is a resubmission of an earlier submission. The following is a list of the peer review reports and author responses from that submission.

Round 1

Reviewer 1 Report

Comments and Suggestions for Authors

Dear Editor

WATER

Hello

Thank you very much for the opportunity given to me to review the attached manuscript. After a careful reading of the work hinging on the assessment of ecological stability in four snaps for a watershed in Slovakia in connection with soil erosion changes resulting from changes in land-use and precipitation, I found it an immature manuscript in different aspects in the present form, despite the hard work conducted for the research. The work from the viewpoint of the subject and goal was potentially a good one, and I buy it. It gives some information that would be interesting to the journal's readership, but there was no rationale for the work.

- The present gap in the study field has not been highlighted specifically to verify the necessity and novelty of the present work. It has to be properly justified why the work is needed. 

- Despite the title of the manuscript and the importance of the work, it is not a comprehensive work standing on reliable logic and rationale on the selection of the tillage and study procedures. 

- The introduction is very general, sparse, divergent, non-documented, and no new things were addressed. Chronologically based and convergent assessment and review are needed. No review of literature and state-of-the-art of the existing problems in the study field with further focuses on international and recent papers in the same field has been made. The entire introduction and mainly reviewing of the literature are dispersed and scattered. No convergent style leading to the final goals of the work is seen. Very relevant works of literature are missed. 

- The entire results and discussion would be reviewed after getting assured about the methodology and study procedure!! 

- The practical use of the research findings has to be highlighted at the end of the manuscript in the conclusion. The translation of research results to practical purposes is a vital task for the researcher to be deeply and explicitly considered. I am not sure that such a type of tillage with potential issues dealing with soil compaction and huge manipulation of the soil can be welcome by the users and policymakers.

- Conclusion is not also concise and research specific. It has not provided the necessary information for the readership to lead them to proper and adaptive studies in the future. It has to contain a concise summary of the key findings of the study. Furthermore, it is not necessary to summarize the study or to discuss the methodology. Further, general managerial implications and future research can be briefly pointed out in the conclusion.

Overall, despite the valuable attempts of the authors, it suffers some flaws and also serious challenges and deficits, as annotated in the reviewed manuscript and briefly mentioned above, which makes me against its acceptance for publication in the present form. A potential good work has not been presented properly and well. However, a resubmission of a substantial and fully restructured version as a new manuscript for further review and after insuring incorporation of all comments and suggestions is recommended.  

Best Wishes

Comments on the Quality of English Language

Please see the reviewed file.

Author Response

Thank you for your comments. The changes have been made to the article based on your comments.

Reviewer 2 Report

Comments and Suggestions for Authors

The study of soil erosion intensity from the stability of ecological landscape is relatively novel, which has good practical significance. In particular, based on different evaluation methods, the relationship between different ecological stability indicators (ecological landscape structure) in 1990,2006,2012,2018 and rainfall is discussed.

Questions and Suggestions:

1. Rainfall is the driver of soil erosion, and erosion rainfall is the fundamental driving force of soil erosion. Is it reasonable to explain the special situation of soil erosion in 2012, please give a detailed explanation.

2. Is the conclusion reasonable, the relationship between ecological stability and soil erosion intensity by four years of data (the conclusion of line 451). Whether the data can be added to reflect the relationship.

3. In the discussion section, it is suggested to increase the content of land use structure optimization measures.

Author Response

Thank you for your comments. You can find the answers in the attachment.
